# Void Avoiding Opportunistic Routing Protocols for Underwater Wireless Sensor Networks: A Survey

**DOI:** 10.3390/s22239525

**Published:** 2022-12-06

**Authors:** Rogaia Mhemed, William Phillips, Frank Comeau, Nauman Aslam

**Affiliations:** 1Department of Engineering Mathematics and Internetworking, Dalhousie University, Halifax, NS B3H 4R2, Canada; 2Department of Engineering, St. Francis Xavier University, Antigonish, NS B2G 2W5, Canada; 3School of Computing Engineering and Information Sciences, Northumbria University, Newcastle upon Tyne NE1 8ST, UK

**Keywords:** void avoiding, opportunistic routing (OR), underwater wireless sensor networks (UWSNs), void area, routing

## Abstract

One of the most challenging issues in the routing protocols for underwater wireless sensor networks (UWSNs) is the occurrence of void areas (communication void). That is, when void areas are present, the data packets could be trapped in a sensor node and cannot be sent further to reach the sink(s) due to the features of the UWSNs environment and/or the configuration of the network itself. Opportunistic routing (OR) is an innovative prototype in routing for UWSNs. In routing protocols employing the OR technique, the most suitable sensor node according to the criteria adopted by the protocol rules will be elected as a next-hop forwarder node to forward the data packets first. This routing method takes advantage of the broadcast nature of wireless sensor networks. OR has made a noticeable improvement in the sensor networks’ performance in terms of efficiency, throughput, and reliability. Several routing protocols that utilize OR in UWSNs have been proposed to extend the lifetime of the network and maintain its connectivity by addressing void areas. In addition, a number of survey papers were presented in routing protocols with different points of approach. Our paper focuses on reviewing void avoiding OR protocols. In this paper, we briefly present the basic concept of OR and its building blocks. We also indicate the concept of the void area and list the reasons that could lead to its occurrence, as well as reviewing the state-of-the-art OR protocols proposed for this challenging area and presenting their strengths and weaknesses.

## 1. Introduction

With a large area of the earth (more than 2/3) covered by water [1,2], investigating the underwater environment and exploiting the UWSNs in various areas of underwater studies have become imperative due to the increasing human requirements and needs. Applications in different human activities in the underwater environment have become very important and opened a new field for investigators interested in this area. Many researchers such as [3,4,5,6,7,8,9,10,11,12] have proposed solutions to fulfill human requirements and needs in such a harsh environment for industry (detecting chemical pollution, pipeline monitoring, biological phenomena, and seismic studies), government (military applications and maintaining the coast), and nature (hazard events, marine farms, ecological monitoring, and contamination studies). The underwater sensor design in such applications ranges from simple to complex [13]. However, this UWSN research area is very challenging, and most work that has been conducted using terrestrial wireless sensor networks (TWSNs) cannot be directly implemented into UWSNs because different communication channels are used and the characteristics of underwater environments are unique [3,14,15,16]. Moreover, underwater sensor nodes are expensive battery devices, and they need better protection of their hardware to resist the water characteristics [17,18,19].

One of the main tasks that faces researchers in the networks is determining how to route the information collected by sensor nodes to reach the sink(s) while minimizing routing costs (minimum delay, energy cost, number of hops, and shortest path) and ensuring network connectivity (void area problem). That is, to assure the network continues to function for as long as possible. OR is one of the routing techniques used to transmit data in UWSNs. It is an emerging routing technology that was proposed to overcome the drawback of unreliable transmission, especially in UWSNs. OR uses the broadcast nature of wireless communication to forward data packets to reach the sink(s) through one-hop or multi-hops. It addresses the major challenges of UWSNs, such as energy efficiency, void avoidance, reliability, and network stability. This OR approach takes into consideration the limited resources (battery and memory) of the underwater sensor nodes, and over the years, many forwarder methods in OR were proposed to prolong the network lifetime and increase the chance that every node has a direct or indirect link to the sink(s). Therefore, OR extends the lifetime of the UWSNs. However, based on previous research, it has been determined that there is still space for development in this area [20].

Many survey papers for UWSNs have been published, such as [21,22,23,24,25,26] which cover different directions of studies in the UWSN area, and [2,27,28,29,30,31,32] summarize existing UWSN routing protocols. However, in order to present a more specific survey paper from the perspective of routing, we aim through this survey paper to collect and present a comprehensive overview of the state-of-the-art of routing protocols for UWSNs that focuses on addressing the void area problem by utilizing OR. We also predict future trends and challenges that remain unexplored in order to bring these to researchers’ attention.

The main contributions of this survey paper are:We explain how the void area problem in UWSNs can be addressed by utilizing the OR technique, and we review up-to-date OR protocols. To the best of our knowledge, our survey paper is the first one that reviews up-to-date void-avoidance OR protocols for UWSNs.We classify up-to-date void avoiding OR protocols for UWSNs based on their most important characteristics and features.We identify some of the open issues and challenges in UWSNs, which can assist the designers of OR protocols for UWSNs.

The rest of this paper is structured as follows: Section 2 presents the routing protocols in general. It includes the main challenges facing UWSN routing protocol designers, the void area problem, and the reasons that may lead to its existence in the network architecture. It also includes the concept of OR, the main components, and the classification of the OR protocols based on their construction blocks. A review of the state-of-the-art of routing protocols for UWSNs related to our paper is presented in Section 3. The comparison study between the reviewed protocols, including their architectural features, benefits, and drawbacks, is presented in Section 4. Future challenges to be faced in this area are reported in Section 5. Finally, we conclude this paper in Section 6.

## 2. Routing Protocols

In general, the underwater sensor nodes are deployed in the area of interest following one of the underwater network architectures (i.e., 1, 2, 3, or 4 dimensional UWSNs, which are presented in many papers such as [16,26,32,33,34,35]). The deployed underwater sensor nodes must be organized in such a way that they cover the entire area of interest in order to gather the data whenever an event occurs. Routing protocols are responsible for discovering and maintaining transmission routes. Thus, a route between sensor nodes and the sink needs to be established for effective and reliable data transmission. Routing is the backbone of any network. The sensor nodes can communicate with the sink(s) either by: (1) direct link, where the data packets can be sent directly from the source node towards the sink(s). or (2) through a multi-hop path where the data packets are forwarded by the relay nodes until they reach the sink(s). However, multi-hop communication suffers from the complexity of establishing a route, which has effects on network performance such as capacity, reliability, and efficiency.

### 2.1. Main Challenges Facing UWSN Routing Protocol Designers

In this context, to better design an OR routing protocol for UWSN, a number of challenges that UWSN routing protocol designers encounter are listed and briefly discussed below, as in [36]:Limited bandwidth and data rate: UWSNs suffer from limited available bandwidth (i.e., acoustic waves use a frequency between a few Hz and tens of kHz) and low data rate (i.e., the transmission rate hardly exceeds 100 kbps). The limited accessible acoustic bandwidth depends on the communication range and acoustic frequency.High propagation delay: The UWSNs use an acoustic channel for communication between the underwater sensor nodes themselves and with the sink(s). In the acoustic channel, the propagation speed is five orders of magnitude lower than in the radio channel. This high propagation delay (0.67 s/km) can significantly decrease the throughput of the network.High noise and interference: Basically, there are two kinds of noises, man-made and natural. These noises are caused by water currents, machines, marine mammals, and shipping. The noise under water is much more serious than in the terrestrial environment. The interference is essentially caused by the surface, the bottom, or animals and contamination reflections.High bit error: Due to the shadow zones caused by animals, water currents, and human-made noise, the acoustic channel suffers from a high bit error rate and temporary losses of connectivity.Limited resources: In UWSNs, sensor nodes are constricted resource devices (i.e., they have limited energy and memory). Therefore, after deploying the sensors in an underwater environment, it becomes difficult and costly to replace or recharge the node batteries due to the harsh underwater environment. Moreover, underwater sensor nodes are vulnerable to deterioration and damage due to corrosion and pollution.Topology changes: Due to the flow of water, the underwater sensor nodes cannot stay in one location; instead, they move randomly, which gives UWSNs a mobile or changeable topology.

### 2.2. Void Area Problem in UWSNs

Sensor nodes drift at different depths in the commonly deployed three-dimensional UWSN architecture to make it easier to identify or monitor a certain phenomenon. Multi-hop routing protocols depend on the relay nodes with positive advancement to transmit the data collected from the phenomenon from the source node to reach the target location on the water surface (sink(s)), as illustrated in Figure 1. In this figure, data can be transmitted through the path with dotted arrows to reach the middle sink or through the path with solid arrows to reach the sink on the right side. One of the very critical issues that face data transmission, particularly with UWSNs, is known as the void area problem as it appears in Figure 1. The data will be stuck in the relay node in the dotted path since the upper hemisphere area of the relay node is empty and there is no other node closer to the sink than this relay node. Numerous researchers have recently become more interested in this issue; however, much more research is required before it can be fully addressed. In this paper, we will follow the same UWSN routing protocol classification, location based and location-free based categories, as presented in previous works [15,25,29,31,37] in order to give a clear understanding of void area characterization. In location-based routing protocols, the detected data is transmitted from the source through the relay nodes with a shorter Euclidean distance to the destination on the sea surface. The upward region of the node’s sphere is referred to as the void area if, during this procedure, any node that is holding sensed information could not locate a relay node in its communication range with a shorter distance to the destination to transmit the information to it. In this case, the node is known as a void node. In the location-free based routing protocols, the sensed data is forwarded by sensor nodes using their depths until it is delivered to the sea surface. The sensed data is transmitted from the source through low-depth relay nodes to the surface. In this category of routing protocols, a node holding information is referred to as a void node and the space above it as a void region if it is unable to connect with a node that is in its transmission range but has a lesser depth than itself.

Therefore, a communication void or void area between underwater nodes exists when the area has an absence of nodes. The void area is one of the essential problems to investigate in the UWSN field. It can prevent communication between two or more network sensor nodes, which in turn can lead to a topological partition that results in decreased network connectivity and increased packet loss, which lowers the performance of the entire network.

Through our research and review of the literature, we have come to conclude that any network architecture can experience the void area phenomenon due to one or more of the following causes [36]:Sparse topology deployment: Because underwater sensor nodes are expensive, it may not be possible to deploy enough of them to cover the area of interest. These sparse networks are prone to empty areas.Underwater sensors failure: Owing to the harshness and peculiarities of the underwater environment, it is more likely that the sensors would malfunction due to corrosion and fouling, which could result in a void area issue.Movement of the underwater sensor nodes: Water currents cause the underwater sensor nodes to move in both horizontal and vertical directions. Both node placements and the network topology will change because of this relocation, and a void area might be created.Temporary obstacles: Many living organisms are found in the underwater environment. The movement of organisms could interfere with the underwater sensor nodes’ ability to communicate. In addition, ships, boats, and other water-surface machinery might block the communication link between the network devices. As a result, void areas could be created.The acoustic channel characteristics: The underwater environment characteristics have an impact on the acoustic communication channel, changing the signal’s quality and strength at different water depths due to disturbed pressure, temperature, and salinity at varying water levels.

### 2.3. Opportunistic Routing

At the beginning of this section, we will give the definitions of some terminologies that are used in this paper to make it simple and clear.

Neighbor nodes/neighboring nodes of node (i): are a set of nodes, which are in the transmission range of the node (i).

The qualified set: is a subset of the node’s neighbor, which meet the rules adopted by the author(s) to design an efficient routing protocol.

Next-hop forwarder of node (i): the nodes within the transmission range of the node (i) and at the same time have depths less than the depth of node (i).

Beacon messages: Status messages that contain the status information of the nodes, which are used to exchange this information between the neighbors.

P_holder_: is the node carrying the data packet in the current round. It could be the source node, which originally generated the data pocket, or a relay node.

Relay nodes (i): any node located between source node and sink(s) that carries on forwarding the data packet starting from the source hop by hop until it reaches the sink(s).

OR is a promising technique that was proposed for overcoming acoustic signal fading, high bit errors and losses due to shadow zones, limited bandwidth, high power consumption, and signal spreading [38]. The main concept of OR is to use the broadcast nature of wireless networks, which allows multiple nodes to overhear the transmissions made by any in-range sensor node. Therefore, various underwater OR protocols have been suggested in order to enhance communication in underwater networks.

In OR protocols, a subset of a node’s neighbors will be selected as next-hop forwarder set candidates. These nodes collaborate in a coordinated manner to continue forwarding the packet along toward the destination (sink) by using a prioritized technique according to the rules implemented by the protocol [1,14,39]. The forwarding candidate set selection and the coordination manner between these forwarding candidates to deliver the packet are the two main parts of OR construction. This OR approach is preferable to the traditional multi-hop routing approach, in which only a single node is selected to act as a next-hop forwarder, to increase the probability of delivering the packet [1,25,30]. This can be illustrated through the following example:

Let us assume that the delivery probability of each link (which presented by the arrow in Figure 2 is *p* and (0< *p* ≤ 1).

In the traditional multi-hop routing approach, the delivery probability, *D_Prob,,_* from the source node to the sink using *h* hops can be presented mathematically as.
(1)$$D_{Prob}=p^{h}$$

In contrast, if all the relay nodes can transmit the packet by using the OR approach, the probability of delivering the packet to the sink is increased, as explained in [40]. For OR with m possible relay nodes in each hop, as shown in Figure 2, we can express the *D_Prob_* mathematically as
(2)$$D_{Prob}={\left(1-{\left(1-p\right)}^{m}\right)}^{h}$$
where *h* is the number of hops between the node that originally generated the packet and the final sink, and *m* is the number of relay nodes in each hop.

Consider the following numerical example: assume *p* = 0.8, *m* = 3, and *h* = 4. By using Equation (1), the delivery probability is 0.4096 for the traditional routing, while by using Equation (2), we get a delivery probability of 0.9684 for the OR routing. Figure 2 below illustrates both routing protocols.

Hence, by taking into account the advantage of the broadcast nature of the wireless transmission medium [41] and using the OR forwarding technique, it has become possible to mitigate the effects of the underwater environment and its characteristics on the acoustic communication channel and improve the efficiency of the underwater acoustic physical links [1,25,42]. That is, the OR technique has been proposed to enhance network performance by reducing high bit errors and losses caused by limited bandwidth, high power consumption, and signal spreading [38]. Moreover, using OR reduces packet retransmission; retransmission will only take place when none of the next-hop forwarder set candidates receive that packet. Taking into account OR features, a number of OR protocols for UWSNs have been developed in recent years. These OR protocols utilize multicast mode, in which a single source node transmits its data to multiple nodes by utilizing more than one link at the same time to form the next forwarder candidate set.

#### 2.3.1. OR Construction Blocks

The OR protocol technique is essentially constructed on two important building blocks, as illustrated with their classifications in Figure 3. These building blocks are candidate forwarding set selection and candidate set coordination [1,21,25].

##### Candidate Forwarding Selection

The first building block in OR protocol design is the candidate forwarding set selection process. Selecting a subset of nodes from the source’s neighboring nodes to be the qualified set to carry on the packet and continue the forwarding procedure is the responsibility of this process. More generally, based on the next-hop forwarder node-selecting technique, the candidate forwarding set selection procedures can be classified into the three following categories [1,25]:Sender-side-based candidate set selection: in this category, beacon messages between the nodes in the networks are used to exchange the information of the sensor nodes and make it available within their neighborhood. The current forwarder node, which has a data packet to transmit, will use this information to facilitate its mission to determine its next-hop forwarder candidate set.Receiver-side-based candidate set selection: in contrast to the first category, this one requires the neighbors to check the data packet’s header when they receive a data packet from the sender in order to identify which received nodes are eligible to be candidate nodes and which ones are not. In this category, each neighboring node is responsible for determining whether it will be included in the list of the potential next-hop forwarder candidate set or not.Hybrid candidate set selection: In this category, the next-hop forwarder candidate set is determined cooperatively by the current forwarder node, which has the data packet to transmit, and its neighbor nodes by exchanging their information.

##### Candidate Set Coordination

The coordination phase is the second and most significant building block in constructing an OR protocol. In order to continue forwarding the data packet until it reaches its destination, the nodes in the next-hop forwarder candidate set must cooperate in a coordinated manner. According to the protocol’s regulations, the node with the highest priority (i.e., the most suitable node) will transmit the packet in this case, deferring transmission to other candidates with lesser priorities. If the node with the higher priority cannot finish its transmission, the node with the next higher priority will begin its transmission, and so on, until the packet reaches its destination.

By functioning in a coordinated manner, this building process supports enhancing the network’s throughput and the routing protocol’s accuracy by preventing packet duplication. Packet duplication causes unnecessary and redundant transmissions, wasting the node’s energy. Additionally, the overall collision rate can be decreased.

The coordination procedures between the candidate nodes can be divided into the two following categories [1,25]:Timer-based candidate set coordination: Each candidate node in this process has a holding time based on its priority. As a result, the candidate keeps the source’s received data packet in their possession for a while (the holding period). The remaining candidates will suppress their transmission if the highest-priority node successfully transmits the packet and if they get an indication during the waiting period. If not, the packet will begin to be forwarded by the node with the next highest priority when its holding time expires, and so on.Control packet-based candidate set coordination: The candidate nodes in this approach communicate with one another by exchanging control packets. Therefore, a candidate node responds to a packet with a brief control message. This control packet transmission is used to notify the currently active forwarder node that the packet has been successfully received. It also notifies the other low priority candidate nodes to pause their transmissions.

#### 2.3.2. OR Classification

The existing OR protocols in UWSNs can be classified based on their positioning information into two main classifications: geography-based and pressure-based. In the first category (geographic-based), selecting the forwarding set candidates and making the forwarding packet decisions in OR requires information about the geographic location of sensor nodes. While in the pressure-based category the depth information of nodes is needed to select the next forwarding set candidates and make the forwarding packets decisions. This classification with the state-of-the-art reviewed protocols can be seen in Figure 4.

## 3. Review on Opportunistic Routing Void Avoidance Protocols for UWSNs

Only a few protocols have been proposed to deal with the void communication area problem in UWSNs using the opportunistic routing technique. In this section, we will give a quick review of all the existing protocols.


**
HydroCast
**


Authors in [43] presented a hydraulic pressure routing for underwater sensor networks protocol (HydroCast). HydroCast forms a cluster of nodes by using only the local knowledge of the topology, excluding hidden terminals among them, while also maximiing the expected packet advance (EPA) of this cluster. When adopting the time of arrival technique, which is frequently used in UWSNs, the current forwarder node in HydroCast can define the pairwise distances and two-hop connections for the nearby nodes in order to determine its forwarding set. Additionally, the forwarding set candidates are prioritised using a distance-based timer approach. To help organise the transmission and reduce collisions, when nodes in the forwarding set receive a data packet from a recent forwarder node, they set their timers in order, starting with the node with the longest distance.

HydroCast also proposes a *Local Lower-Depth-First Recovery* approach and *2-D Void Floor Surface Flooding for Recovery Path Search* for a recovery mode. Where each void node (i.e., local minimum node as used in the paper) seeks out its neighbors to find a node with a lesser depth than itself, this lesser depth node could be another void node with a new recovery path or a sensor node in a position that helps to resume the greedy forwarding techniques. Figure 5 shows the recovery path in the HydroCast protocol.

In the 3D network topology, nodes experiencing a void area employ a costly flooding technique to learn which nodes are best suited to resume greedy forwarding or identify alternative routes to better forwarding channels. However, it is difficult to estimate the limited 3D flooding probability value because the flooding could involve every sensor node and affect the entire network topology. They suggest 2D flooding on the surface of the void floor to get around this restriction and increase the effectiveness of the procedure. This flood will include the best possible collection of nodes. As a result, nodes on the surface will monitor their void floor surface status using their local connectivity information and forward packets accordingly, whereas nodes that are not on the surface but are controlled by surface neighbors will not forward packets.

HydroCast addresses the void area issue using an OR approach, which also successfully enables increasing the packet delivery ratio with small end-to-end delays since a subset of the neighboring nodes simultaneously receive the data packet appropriately. However, at the same time, as a result of using opportunistic routing, the HydroCast protocol suffers from redundant packet transmission, where a data packet may be delivered to the sink multiple times, causing the depletion of network resources. In addition, in terms of energy efficiency, implementing the recovery mode results in additional energy costs. Moreover, there is no evidence provided about the energy consumed by the pressure sensor in order to find its depth.


**
VAPR
**


Void-Aware Pressure Routing (VAPR) [44] is an anycast soft-state routing protocol. It was proposed in order to address the void node issue in UWSNs. VAPR is built up of two stages: the enhanced beaconing stage and the opportunistic data forwarding stage. Instead of falling into a void area and then implementing a recovery mode, VAPR takes advantage of the geographic routing and employs the regular beaconing messages method, which includes some useful local information about the node, in the forwarding set selection stage.

In VAPR, any node that receives a beaconing message from a neighbor updates its neighboring table and examines its depth with the received depth information. The node then makes its own routing decision by removing void nodes (dead ends or local maxima) from its forwarding sets and chooses the overall best route to the destination as shown in Figure 6; this will help avoid the packet from falling into a void area in the network. In fact, implementing the VAPR protocol will prevent data packets from being stuck in a node because the protocol relies on the surface station and the beaconing message sent from it to the sensor nodes below as well as the stored information in the nodes.


**
GEDAR
**


In [45], the proposed protocol, geographic and opportunistic routing with depth adjustment-based topology control for communication recovery (GEDAR), utilizes the greedy forwarding technique by knowing the position information of each current forwarding node, its neighbors, and the known sink. GEDAR follows the sender-side OR category, where the forwarding set candidates are determined in each hop by the sender node. Initially, GEDAR uses a greedy, opportunistic forwarding mode to route the packets. Once a node has gathered some data and needs to transmit these data to a sink(s) node, it includes IDs of its forwarding set candidates in the data packet header and broadcasts the packet to its neighbors. When a neighbor node receives the transmitted packet, it checks whether its ID is in the packet header or not. If it is not a forwarder candidate node, it just drops the packet. Otherwise, it calculates the holding time to decide when it can transmit the packet. This procedure will continue until the packet reaches the sink(s) on the water’s surface. If the packet is trapped in a void node, the recovery mode is applied by GEDAR. In the recovery mode, when the packet gets stuck in a void node (node v in Figure 7), the protocol deals with the problem by taking advantage of a network topology control strategy where any node in a void area can move in a vertical direction (from *D_1_* to *D_2_*) to adjust its depth. Then it bypasses the void area to be able to communicate with other nodes trying to resume the greedy forwarding. Therefore, the void node first discontinues sending the gathered packets and starts calculating a new depth that will allow it to continue its OR greedy forwarding to deliver the data packet to the next hop.

The recovery technique used by GEDAR helps by bypassing the void area, which as a result, improves the networks connectivity and increases the packet delivery ratio. On the other hand, in energy consumption terms, this Depth Adjustment technique implemented by GEDAR exhausts a very high amount of energy in physical movement to adjust the network topology, and this will make nodes exhaust their energy rapidly and reduce the network lifetime.


**
IVAR
**


An Inherently Void Avoidance Routing Protocol for Underwater Sensor Networks (IVAR) [46] is a receiver-based forwarding protocol, so the forwarding node does not need to store its neighbor’s information. In IVAR, a hop-by-hop forwarding set selection technique is used to forward the data packets from the sensed node to the sink. Each packet holder uses local information about hop distance and packet advancement to determine its own forwarding set, and the nodes in these forwarding sets are arranged and given a priority depending on two metrics: their hop count as a first metric and their depth as a second one, to forward the packets. IVAR uses beaconing messages sent from the destination to the source; this helps the sensor nodes get the reachable information of the sink(s) and relay nodes. Therefore, the void nodes (yellow and red nodes), as Figure 8 shows, will be excluded from the forwarding set of the sensor node, and the route with a lower hop count will be chosen.

Choosing a route with a lower hop count manages the energy consumption and reduces the packet delivery time. Besides, using the node’s depth assists in preventing packet duplication. On the other hand, because of the broadcast nature of the protocol and because the qualified forwarding nodes may be distributed around the forwarding node in various directions, the protocol cannot completely suppress route and transmission duplication. This limitation will cause the hidden terminal problem and, consequently, extra energy consumption. IVAR uses a periodic beacon by the sink to update the underwater nodes with their current position in the network. Therefore, all the routes from the sink to the sensor nodes will be established in advance, and all the routes directing the packets to void areas will be excluded. However, the beacon interval has to be chosen cleverly because it has a great effect on node information and communication efficiency, which consequently will impact network performance.


**
OVAR
**


The opportunistic void avoidance routing (OVAR) protocol [47] is a sender-side method and a soft-state routing protocol, that requires some local reachability information (e.g., hop count distance, forwarding direction, etc.) about one-hop neighbors to be held in every node. This provides a general observation of each node on the topology. OVAR was proposed to handle IVAR weaknesses (i.e., hidden terminal problems and duplicated packets of transmission). In the same way as IVAR, to handle the problem of void areas, OVAR implements the beaconing procedure and considers its benefits. Different from the receiver side IVAR protocol, in the sender side OVAR protocol, the one-hop neighboring information is held in the sensor node to establish an adjacency graph at each forwarding node. In terms of energy consumption management, OVAR deals with the number of nodes in the forwarding sets, where the size of the forwarding set can be adjusted based on the network density to save energy by reducing the energy consumed by a node when it receives a packet. Reducing the forwarding set size may reduce the delivery ratio and increase packet retransmission, which will lead to more energy consumption. In terms of void area, OVAR includes the high-depth nodes in the forwarding set, which may be inefficient in terms of reliability, energy consumption, and protocol latency. OVAR is slightly more complicated than IVAR, which is caused by the procedure OVAR implemented to eliminate the hidden nodes problem and its effects on the protocol’s performance, in addition to the trade-off procedure between energy consumption and reliability.


**
VHGOR
**


Void handling using geo-opportunistic routing in underwater wireless sensor networks (VHGOR) [48] adopts geography-based opportunistic routing (GOR) to forward data packets to reach the destination over multi-hops. It is a heuristic protocol implemented using two metrics to form optimal forwarder selection. OREPP metrics try to positively advance the data packets towards their destination. The first metric is opportunistic routing based expected packet progress (OREPP), which is calculated based on the difference between the geographic distance between the source and destination and the geographic distance between any node and the destination, residual energy, and packet delivery probability. The second metric is node closer to the destination (NCD); NCD can be defined as the best node with maximum OREPP to forward the current packet. VHGOR uses a greedy forwarding approach to advance the packet through each hop towards the destination, and if the packet becomes stuck in one of the forwarding nodes, then it switches to the void mode. VHGOR handles the void problem using the two following techniques:Convex void handling: If the packet is stuck at the NCD node, VHGOR attempts to identify a different path to forward the same packet to the destination by removing the present forwarder and re-establishing the convex structure with remaining neighbors founded in the neighbor table (NT).Concave void handling or recovery mode: The void becomes concave when a packet gets trapped in a node without any neighbors with lower pressure levels, which means that its NT entry is empty. In order to reroute the packet along a different path to the destination, VHGOR manages the concave void by rerouting it down the recovery path, which runs from downward to upward. The previous sender chooses the subsequent NCD node from its NT to continue sending the same packet after receiving the packet from the concave empty node.

Figure 9 demonstrates the forwarding packet route and recovery mode that VHGOR has adopted. Node n_1_ chooses node n_2_ to be the next forwarding node, since node n_2_ has the highest Expected Packet Progress (EPP) value in its neighbor table (direction number 1 in the figure). In the same manner, node n_2_ chooses node n_3_ as the next forwarding node and transmits the packet to it (direction number 2 in the figure). However, since node n_3_ is a void node and has no nodes to forward to, node n_3_ returns the message back to node n_2_ (direction number 3 in the figure). The next node in Node n_2′_s neighbor table is subsequently chosen as the next forwarder (direction number 4 in the figure). Since D is inside the transmission range of node n_10_, node n_10_ finally delivers the packet to D. In order to create the best forwarder from FCS, VHGOR takes into account residual energy, which helps cut down on energy consumption.

VHGOR considers the residual energy in forming the optimal forwarder from the forwarding candidate set (FCS), which assists in reducing the energy consumption. Besides, employing opportunistic forwarding works in improving the delivery ratio at the same time introduces end-to-end delay to some extent [48].


**
WDFAD-DBR
**


In [49], another pressure-based routing protocol was described in detail, namely the weighting depth and forwarding area division DBR routing protocol (WDFAD-DBR). To increase the reliability of the packet transmission and decrease the probability of the occurrence of a void area, WDFAD-DBR uses the weighting depth difference of two-hop nodes to construct its routing decision. As presented in Figure 10, node S is a source node, and the two forwarding candidate nodes with lesser depth are A and B. In the greedy protocol DBR, node A has a lesser depth than node B, giving A the priority to transmit first. Node B will suppress its transmission and drop the packet when it hears it from node A. However, a void area occurs since there are no nodes in node A’s transmission area (S2) with less depth than node A to carry forward the packet. In contrast, WDFAD-DBR selects node B to forward the packet because it considers both depth differences, current depth difference (node B depth—source depth), and the difference depth of the expected next hop (node E depth—node B depth).

In WDFAD-DBR, the void nodes can remove themselves from the data packet routing to increase the opportunity for the other candidates in the forwarding set to forward the packet. In addition, to control the number of forwarding nodes, WDFAD-DBR divides the forwarding area into a constant primary forwarding area (the Reuleaux triangle) and two auxiliary forwarding areas, which might be extended or shrunk depending on node density and the quality of the channel. In terms of energy consumption, on one one hand, to help reduce the energy expenditure due to the duplicated packet transmission, the auxiliary forwarding area is divided into a number of smaller sub-areas, which helps save some energy. On the other hand, the periodic neighbor requests and the corresponding ACKs in a reply to each control packet exhaust the energy of the nodes and waste network resources. In order to bypass the void area, WDFAD-DBR successfully detects the void nodes and excludes them from the forwarding procedure. However, the protocol fails to detect the trapped nodes in advance. Moreover, when a fixed primary forwarding area is implemented by the protocol, the flexibility of the routing might be restricted in its ability to choose and adjust the forwarding nodes under various conditions.


**
EVA-DBR and SORP
**


In [50,51], the energy-efficient and void avoidance depth-based routing (EVA-DBR) protocol and A Stateless Opportunistic Routing Protocol for Underwater Sensor Networks (SORP) are proposed. SORP builds on the performance evaluation from [50], considering a realistic sensor mobility model, the shadow zone, variable propagation delays, and additional network parameters and results. EVA-BDR and SORP are routing protocols consisting of two phases: the updating phase and the routing phase. The protocols depend on the information broadcasted periodically in the updating phase from the neighbor nodes that are one-hop away from the source node for void detection and bypassing in the routing phase. Initially, all the nodes in the network are homogeneous. However, in the updating phase, the void and trapped nodes are detected over time by the broadcasted information from the neighboring nodes. In addition, through the updating phase, each regular node will choose its best candidate node in terms of the expected packet advancement (EPA) among the neighboring nodes with lesser depth to be used as a reference node in the opportunistic data forwarding [50,51]. In the routing phase, to increase the packet delivery probability in each data transmission operation, all the detected void and trapped nodes take themselves out of the forwarding set; this procedure will increase the opportunity for the other regular nodes in the forwarding set to forward the packet. In addition, the forwarding area can be resized depending on the density of the network, as presented in Figure 11, and all the qualified nodes will set their forwarding timer to forward the data packet. This forwarding time should guarantee a priority-based scheduling of the nodes in the forwarding set and should suppress the duplicate packets.

In terms of energy consumption, since the nodes do not need to send an ACK to the node’s neighbors as a reply to their control packets, the energy consumed per node will be somewhat reduced. In contrast, both protocols may allow the duplicated transmissions to increase the packet delivery probability in a sparse network in addition to periodic broadcasted information exhausting the node’s battery and, as a result, decreasing the network lifetime as well as the node’s life. And in terms of void avoidance, the state of excluded nodes from the forwarding set that announced themselves as void or trapped nodes may change during the transmission data packet or before the period of broadcasting information expires, which may effect the energy consumption and reliability of the network. Moreover, maintaining the neighboring table and the two-hop information will adversely affect the limited resources of the nodes (i.e., energy and memory).


**
EDOVE
**


This section reviews the energy and depth variance-based opportunistic void avoidance (EDOVE) protocol that was presented in [52]. EDOVE was proposed on the basis of the work presented in [24], called the WDFAD-DBR protocol. The protocol addresses the void area problem by selecting the forwarder candidates among the total distributed nodes that have i) a large residual energy and ii) several neighboring nodes within its transmission range (neighbors). Each node in the network architecture shares its information with its 1-hop neighbors using neighbor request and neighbor acknowledgment packets, and each node must keep its neighbor table updated in order to obtain this relevant node information when needed. Once a sender has a data packet to deliver, all of its neighbors will inevitably get it due to the broadcast nature of the protocol. From then, the packet must be transferred through one of these neighbors to the next hop or directly to the destination (sink(s)). In contrast to WDFAD-DBR, EDOVE uses the two-hop depth differences, the normalised residual energy of the node, the next hop depth difference to the source, and the depth difference variance between the neighbors to compute the holding time. This is because the receiving nodes have different residual energies, and EDOVE takes this diversity into account. The holding time parameters are shown in Figure 12.

Finally, to choose the best forwarder node, EDOVE makes the decision by calculating the holding time and selecting the receiving node with the largest residual energy, the greatest depth difference to the source, the greatest depth difference to its neighbor, and many neighbors with a large variance in their depth differences. More factors are taken into account by the protocol, which increases energy efficiency, prevents packet collisions, and extends network lifetime. However, in dense networks or when the size of the network is increased, there are increases in the probability of duplicated packet transmission because the number of nodes with the same depth will increase, making their estimated holding times almost the same. This results in an increase in data packet traffic, which in turn increases energy consumption. Additionally, the protocol views the void area only as a series of energy holes, despite the fact that it serves a variety of purposes, as stated above.


**
TORA
**


The totally opportunistic routing algorithm (TORA) is proposed for UWSNs in [53]. TORA is a novel anycast, receiver-based opportunistic, and geographical routing protocol. It is suggested in order to prevent horizontal transmission, minimize end-to-end delay, address the issue of void nodes, and increase network performance and energy efficiency. The three steps of the proposed protocol’s operation are node localization, candidate forwarder selection, and data transmission.

At the water surface, the multi-sink network architecture is installed, and ordinary nodes drift in different levels underwater, as shown in Figure 13. The ordinary nodes are divided into two types: 1) single transmission node (STN) that are in transmission range of surface sinks; and 2) double transmission node (DTN) that are not within transmission range of surface sinks, they estimate their position by communicating with STNs.

To locate nodes in the network, the time of arrival (TOA) and range are used. Sinks periodically send hello messages to help collect node information that will be used to determine ordinary node location in the localization phase. Next, based on the nodes’ geographic coordinates and remaining energy, the best forwarding node that has a higher residual energy and is closer to one of the sinks will get a higher priority to relay the packet in the candidate forwarder selection phase. After that, the data transmission phase starts once a node has a data packet ready to send. This data packet should be delivered to one of the sinks in a multi-hop fashion through selected forwarding relay nodes. TORA utilizes 2-hop Ack to make sure that the packet has traveled for two hops and zero Acks to reduce end-to-end delay and retransmissions. As a conclusion, data is transmitted to the sink node using a combination of several short, active links.


**
EBER^2^
**


In [54], an energy-efficient and reliable protocol called an energy balanced efficient and reliable routing protocol (EBER^2^) has been proposed to address the void areas. EBER^2^ adopts the potential forwarding nodes (PFN) concept to tackle WDFAD-DBR shortcomings. Since WDFAD-DBR experiences void area problems in some cases because it ignores taking into account the PFNs for the second hop, it suffers from high duplicate packets and collisions, which reduce protocol performance and efficiency. In EBER^2^, the network architecture consists of three types of sensor nodes (sink nodes, anchored nodes, and relay nodes), as demonstrated in Figure 14.

The authors of EBER^2^ take into account three factors as primary parameters for choosing the next forwarder in order to address the WDFAD-DBR weaknesses. The first parameter is the weighting depth difference of two hops; by choosing the next forwarder node based on the depths of the first two hops, the likelihood of a network void area problem is reduced. The second factor is the number of PFNs, which are nodes that are within the source node’s upper hemisphere of its transmission range. A void node is one that has no PFNs; as a result, it is excluded from the upcoming forwarding set, which improves network stability. The residual energy is the third parameter, and it is used to provide PFNs with the same depth but varying holding durations in order to prevent duplicate packets. In addition to forming the next forwarder set and preventing void nodes from being chosen as candidates for the next forwarder, these three parameters also support energy efficiency, boost packet delivery ratio, and lengthen network lifetime by preventing duplicate packets and the ensuing collisions.

To further assist these nodes in communicating with the embedded sinks and delivering data packets to them rather than travelling through a long path to reach the sinks on the surface, the EBER^2^ protocol deploys two additional embedded sinks in the underwater area of interest that have high traffic density, as can be seen from Figure 15. In general, since the nodes placed in these dense and high traffic areas transfer the received packet to the closest embedded sink rather than transmitting further to the surface, this strategy enhances network packet delivery ratio while consuming less energy. Instead, the cost of communication rises because of the high-speed optical fiber links used to connect embedded sinks and on-surface sinks. Additionally, EBER^2^ employs a transmission energy adaptation mechanism that enables nodes that are closer to sinks to reduce their transmission power level in accordance with their distance from that sink. This minimizes the void area created by the death of these nodes by preventing the nodes close to the sinks from rapidly exhausting their energy due to being involved in the majority of forwarding procedures.


**
EDORQ
**


The authors of [55] proposed a new receiver side-based routing protocol for UWSNs named Energy-efficient Depth-based Opportunistic Routing with Q-Learning (EDORQ). The EDORQ contains two phases: 1) the candidate set selection phase to choose a subset of neighbor nodes to carry on forwarding data packets until delivered to the destination; 2) the candidate set coordination phase, where the candidate nodes collaborate according to their priorities by applying the timer-based mechanism to suppress redundant forwarding. Moreover, the authors adopted the Q-learning technique to design the holding time of the candidate nodes. By defining a holding time for each candidate, the candidate node with the larger Q-value has a higher priority, a lower holding time, and will transmit the packet first.

EDORQ starts the forwarding process using greedy mode, where the current forwarding node broadcasts the data packet to its neighbors. Each candidate neighbor extracted the depth *(d)* and void-flag information of the current node from the packet header after receiving the data packet and then compared *d* with its own depth. In order to ensure that the data packets are quickly sent in the sink’s direction, the greedy mode helps locate a collection of candidate nodes closer to the water’s surface. In order to achieve this, the void-flag field in the packet header is set to “0,” indicating that only nodes whose depth is less than the current forwarder are eligible to be chosen as candidates. However, the protocol switches to void recovery mode when the packet is stuck in the void node as, illustrated in Figure 16.

The current node will retransmit the data packet in a void recovery mode, where the value “1” is entered in the void-flag field, allowing the neighbor nodes with the greatest depth to be chosen as candidate nodes. A node should only forward packets with the same ID once for a predetermined period of time in order to reduce duplicate transmissions, comparable to the DBR. As a result, the new candidate set of current nodes would continue the forwarding process. The next best packet forwarder from the current node will then transmit packets in a greedy manner toward the sink if no other the void node is reached.


**
RPSOR
**


In [56], another novel OR protocol called Reliable Path Selection and Opportunistic Routing (RPSOR) for UWSNs is presented to address the void area problem in UWSNs. It is an improved version of the WDFAD-DBR protocol. RPSOR operates in two stages: knowledge acquisition and packet forwarding. In the knowledge acquisition stage, nodes exchange their information through hello packets sent from surface sinks, neighbor request packets, and ACK packets generated by each sensor node. In addition, the node maintains three different tables, which are the source info table, the first hop info table, and the routing table. Furthermore, in the packet forwarding stage, the decision for PFN selection will be made based on the priority function, which is defined by three elements: the reliability index, the advancement factor, and the shortest path index. RPSOR only selects the nodes of the upper hemisphere as the forwarding neighbors. Therefore, nodes having higher a depth than the current node simply drop the packet.

In RPSOR, two sinks are mobile, as can be seen in Figure 17. Mobile sinks are utilized to travel to denser network areas that experience high traffic.

At the beginning of each simulation round, the network uses hello messages to assess the node density at various hops, and it then permits the sink to travel to any hops with a high node density. The nodes located at the following hop must transmit a large amount of load created by denser network locations. The majority of the packets are lost when this high traffic enters the network’s sparse area since the network cannot handle such high traffic levels. Utilizing the position data of the denser hop, which was acquired by the greeting message, the sink determines the vertical trajectory.


**
PCR
**


Recently, a novel power control-based opportunistic (PCR) routing protocol for the Internet of Underwater Things (IoUTs) was proposed in [57]. They develop opportunistic routing and transmission power control methods in order to send data in IoUTs with the least amount of energy possible. Each node in PCR checks many transmission power levels before selecting its candidate set for the next-hop. The PCR protocol implements a periodic beaconing technique during the neighbor discovery phase for each transmission power level in order to gather information from neighbors and update the neighbors table. The candidate set will be expanded to include the neighbor node exhibiting positive packet progress. The appropriate transmission power level and the next-hop forwarding set are then computed based on the energy waste for each candidate set. Hence, the set of candidate nodes with the least energy waste is chosen as the best candidate set to continue forwarding the packet to the next hop until the packet reaches the destination. The nodes in the candidate set will then be sorted based on their normalized packet advancement to define the node’s priority. Then, PCR applies a timer-based approach to manage the transmission coordination between the candidate nodes. Therefore, the candidate node’s packet holding time decreases as its priority increases. Additionally, if a lower priority candidate node detects packet transmission from a higher priority candidate node, it will cancel its own transmission.

By changing the transmission power level at each hop, the PCR packet delivery ratio was enhanced in order to select the most suitable candidate nodes from the sender neighbors to continue passing data packets to the sink(s) on the water’s surface. In dense networks, PCR also lowers the node’s transmission power level to lessen the need for retransmissions, which lowers energy usage in some cases. The energy consumption is still higher than the related works, as we can see from their data, and this will shorten the lifespan of the network.


**
SEEORVA
**


A secure and energy-efficient opportunistic routing protocol with void avoidance for underwater acoustic sensor networks, (SEEORVA) was presented in [58]. This protocol employs the OR strategy for reliable data delivery in UWSNs and uses energy thresholds in the forwarding process to give a priority to the forwarding nodes, which have energy above that particular threshold; in that way, energy efficiency and expanding network lifetime can be achieved. The protocol handles the communication void problem and encrypts transmitted packets using a secure, lightweight encryption technique for security.

In SEEORVA, the best forwarder selection was performed as follows: when a source node has packets to transmit, it creates a virtual vector pipe to the sink (as can be seen in Figure 18). The source then lists all the nodes that are detected within this pipe to be considered, calculating the highest energy of the nodes and the threshold energy value based on the calculated highest energy value.

In the forwarding process, only candidate nodes within the source transmission range that have residual energy greater than the threshold and are making maximum progress to the sink will be given the highest priority and chosen as the best forwarder node. If this node could not forward the packet within the allocated transmission time, the next node in the list would forward the packet to the sink. Therefore, to ensure the security of transmitted data, SEEORVA uses a lightweight security protocol, the novel tiny symmetric encryption algorithm, to encrypt data packets before sending them through the network to the sink. These encrypted data packets can only be decrypted by the collection and processing centers at the water’s surface.

Moreover, the proposed protocol addresses the void problem by encouraging the forwarder node to send a data packet void alert to its previous node if the forwarder node faces a communication void. The previous node searches for an alternative route to avoid the void and uses this alternate route to transmit the remaining data packets from the previous node to the sink.

The nodes’ remaining energy is used as a significant factor to determine the priority of the next forwarder nodes in the forwarding process, thus extending the lifetime of each sensor node and the network overall. While the technique used to handle communication voids gives much better quality of service (QoS) results, it is also easy to implement with less overhead and delay. In addition, the encryption method ensures secure data packet transmission and avoids any leakage in data packets that can be harmful in any way.


**
EEDOR-VA
**


In [59], the most recently published routing protocol to address the void area issue is named energy efficient depth-based opportunistic routing with void avoidance protocol for UWSNs (EEDOR-VA). EEDOR-VA aims to improve network performance by developing a routing protocol that achieves a high packet delivery ratio while using less energy by choosing the shortest routing path. EEDOR-VA decides on routing based on the nodes’ ability to reach the surface sink. This protocol introduces Hop Count Request (HCREQ) and Hop Count Reply (HCREP) messages to update the node’s hop count to the nearest sink that can be approached. In the proposed protocol, data packets will not get stuck in any void and trapped nodes located in the transmission range of a source and/or relay node because these void and trapped nodes do not respond to the HCREQ message and are therefore removed from being one of the forwarding candidates. As a result, each P_holder_ can easily construct its forwarding set. That is, sensor nodes use the information from the hop-count discovery algorithm to update their hop count from the sink(s) and exclude void and trapped nodes in the P_holder_ nodes’ transmission range from being included in the forwarding set. Periodic beaconing and its related costs is eliminated by the hop count discovery mechanism proposed in EEDOR-VA. The main goal of EEDOR-VA is to find as many loop-free paths as possible between a source node and a single or multiple sinks on the sea surface. The protocol can easily change the chosen route from one path to another by electing the next relay nodes from a different path if this relay node is the best choice in the next hop forwarding range. As a result, this technique prevents having to start the hop-count discovery process all over again. If all routes to all of the sinks fail, then a hop-count discovery is initiated. EEDOR-VA updates relay node information using route information and ensures that nodes responding to the P_holder_ have a path to one of the sink(s) in order to avoid the void nodes. The EEDOR-VA protocol’s process is depicted in Figure 19. When a source node has a packet to send, it sends HCREQ first, which is received by all of its neighbors. Each of these neighbors sends out a rebroadcast of the appeal to their own neighbors.

The EEDOR-VA protocol uses rounds; every round is comprised of three phases: a hop-count discovery phase, a forwarding set creation phase, and a data packet forwarding phase.

The hop-count discovery phase is in charge of determining the hop count of any source and/or relay nodes in the network to the sinks, whether the sink(s) are directly reachable within the transmission range of the source or reachable via one or more hops through relay nodes. Once the hop count of each of the route nodes is defined, the forwarding set formation phase is started. In each hop, the P_holder_ forms its next-hop forwarder set based on the extracted candidate information, and only a candidate with a hop count less than the P_holder_ hop count, no matter if it has less or more depth than the P_holder_, will be added to that P_holder_ next-hop forwarder set. Finally, P_holder_ integrates the data packet with the sorted list of the selected forwarding candidate IDs and transmits it to its neighbors. Each neighbor checks the packet header and simply drops the packet if it cannot find its ID in the attached list or starts computing its holding time otherwise.

The EEDOR-VA protocol uses the node’s hop count as the first metric to identify the best forwarding node and the node’s depth as a secondary metric in the event of a tie. The best forwarding node will transmit the data packet immediately after receiving it to continue the forwarding process. Other forwarding candidates will drop the packet if they successfully hear the transmission from the most appropriate node. If not, the data packet will be transmitted by the following node in the sorted list, and so forth. These processes will be repeated hop by hop until the data packet reaches the sink or all the candidate nodes in the forwarding set fail to do so.

## 4. Comparison Study of OR Protocols for UWSNs

In the previous section, the literature review of the state-of-the-art of the OR protocols that are proposed for UWSNs to address the void area problem is presented. The main challenge in the protocols was to handle the void area problem by using different approaches. The occurrence of the void area in the routing path can significantly reduce network performance. In this section, the general comparison of these reviewed protocols based on their characteristics and features is summarized below in Table 1.

In Table 1, we can see that the existing void avoiding OR protocols for UWSNs are classified into two main classifications: geographic-based and pressure-based. In the first category, geographic-based, which includes [44,45,48,57,58], selecting the forwarding set candidates and making the forwarding packet decisions in OR requires information about the geographic position of sensor nodes. While in the pressure-based category, which includes [43,46,47,49,50,51,52,54,59], the depth information of nodes is needed to select forwarding set candidates and make forwarding packet decisions.

The reviewed protocols are divided into sender-side and receiver-side categories based on which node will decide if the candidate node can be added to the next hop forwarder set or not. A higher communication overhead is needed on the sender-side because the sensors frequently need to exchange node information in order to update their neighbors’ tables. As a result, the limited resources of the node (i.e., battery and memory) are used up. On the receiver-side, the sender is unaware of its neighbors and is unaware of its forwarding set. This may result in a significant number of redundant broadcasts and raise the possibility of transmission collisions requiring reiterate transmissions. The entire network stability period may be shortened as a result of packet loss and sensor node energy consumption. A number of these protocols take advantage of the multi-sink architecture and consider the data packet as delivered if it reaches one of the deployed sinks on the water surface. This improves the network reliability.

These state-of-the-art protocols employ various void area handling techniques to address the void area problem in order to increase network performance and deliver data packets properly.

Moreover, Table 1 also includes a brief summary of the benefits and drawbacks of the reviewed protocols in the two last fields. Additionally, most of the protocols deal with the void region by switching from the forwarding approach to the recovery mechanism, and the bulk of them have the stuck node issue. The IVAR, SEEORVA, and EEDOR-VA protocols are the only ones that address the void area issue and recognize every void/trapped node. However, both IVAR and SEEORVA implement periodic beaconing to provide sensor nodes with sink(s) reachability information. The network performance is significantly impacted by the beacon interval. Due to the prioritizing process, which depends on the depth that could be the same for more than one node, and the holding time, which depends on shared parameters between more nodes, both protocols still suffer from duplicate transmissions. While the EEDOR-VA protocol addresses the limitations of these two protocols through the novel hop-count discovery mechanism and the prioritizing technique.

## 5. Open Issues and Challenges in UWSNs

Energy efficiency: Due to the harsh underwater environment restrictions and limitations on recharging or replacing the deployed sensor node, energy efficiency is a major constraint that can restrict many applications from achieving their goals. The current focus of study is on energy conservation and routing process energy optimization. Future studies will continue to focus heavily on this topic.Channel utilization: effective and efficient channel usage is a significant area of investigation in UWSNs that has attracted a lot of attention; it has an effect on energy consumption and void areas that form easily in UWSNs. The channel must be used to its full potential to overcome its limitations, like interference, high error rates, continual sensor node mobility, and propagation latency.Void areas: Due to the reasons listed in Section 2.2, void areas result in more frequent packet drops, decreasing the network QoS. To ensure a high QoS for various UWSN applications and increase network reliability, more mechanisms to handle void areas and void communications effectively and establish trust for user apps are required.Security: A significant area of concern is the security of data exchanged between sensor nodes. The security of data is the most crucial issue in many UWSN applications, especially the military ones. In such applications, any information leak could lead to harmful effects and severe repercussions. To secure the connection between the sensor nodes in UWSNs as attacks and threats grow, investigations that consider security and privacy will be a continuing and extremely difficult effort.

These are some of the most significant and active fields of research for UWSNs that need more investigation, and they will remain so in the upcoming years.

## 6. Conclusions

These days, opportunistic routing in UWSNs has drawn a lot of attention from researchers. OR has been shown to be more effective than the conventional routing strategy for wireless networks because it makes use of the broadcast nature of wireless networks. A number of factors have an impact on the performance and effectiveness of the UWSNs, including a shortage of resources (restricted battery power and memory), the harsh underwater environment, and a weak communication channel. The void area problem employing an OR approach is one of the significant concerns and research challenges in UWSNs. We have investigated in this paper the existing OR protocols proposed to address this problem.

First, we discussed the aspects of routing protocols for UWSNs covering the main challenges facing researchers when designing routing protocols, the concept of the void area problem in UWSNs, and reasons for this problem. OR and its key elements, including OR construction blocks and OR classification have been introduced. Second, the state-of-the-art void avoiding protocols that use the OR technique were investigated in depth. The reviewed protocols have then been compared in many aspects, including the type of protocol, number of sinks, network topology requirements, and the special information required or needed to be maintained during the data packet routing. Their advantages and limitations were listed in the last two columns of Table 1. Moreover, we provided some of the open research issues in UWSNs that require further investigation.

## Figures and Tables

**Figure 1 sensors-22-09525-f001:**
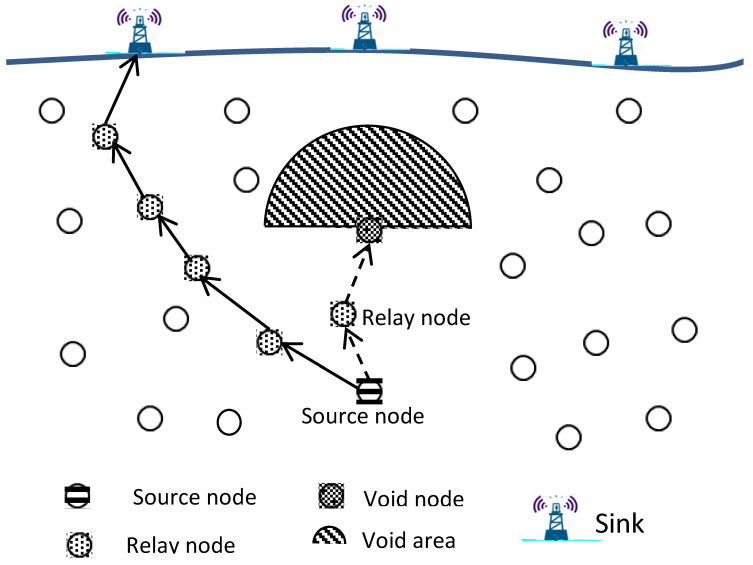
Void area in UWSN architecture.

**Figure 2 sensors-22-09525-f002:**
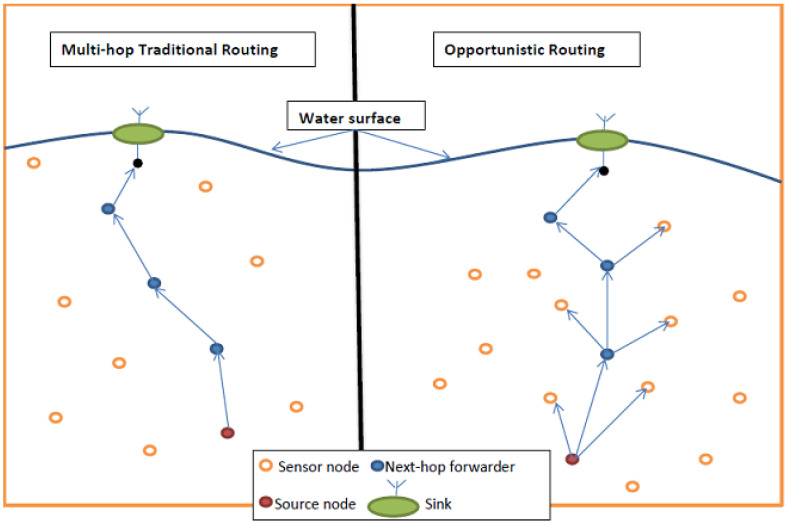
Multi-hop traditional routing vs. OR.

**Figure 3 sensors-22-09525-f003:**
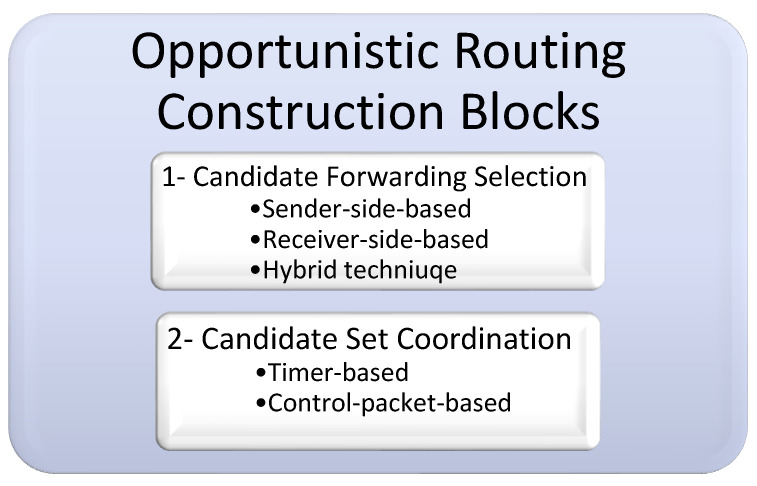
Opportunistic routing building blocks.

**Figure 4 sensors-22-09525-f004:**
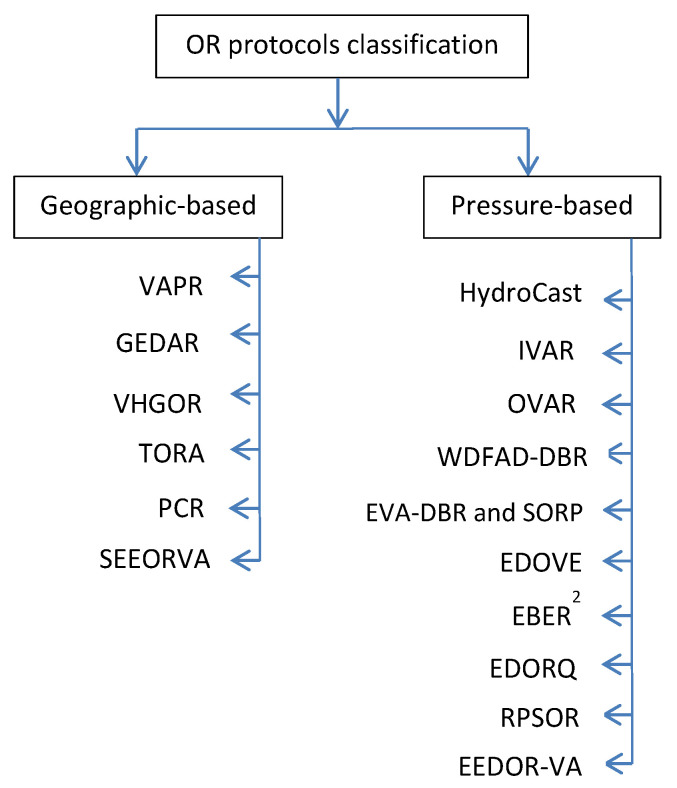
Classification of OR protocols for UWSNs based on position information.

**Figure 5 sensors-22-09525-f005:**
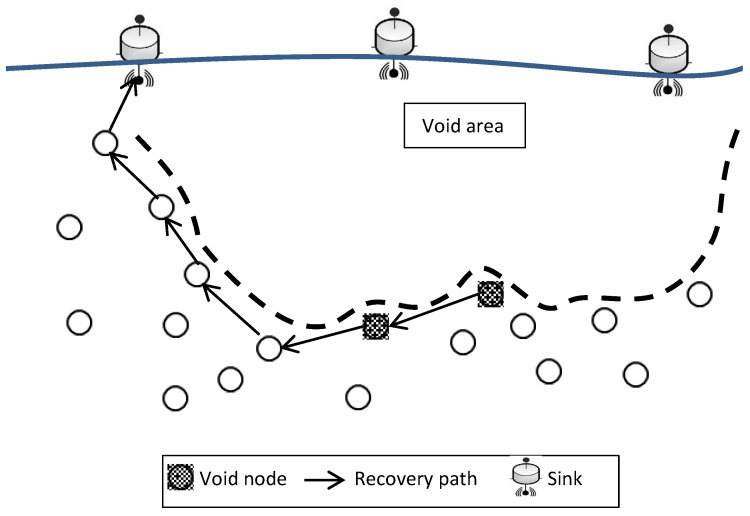
HydroCast void handling technique.

**Figure 6 sensors-22-09525-f006:**
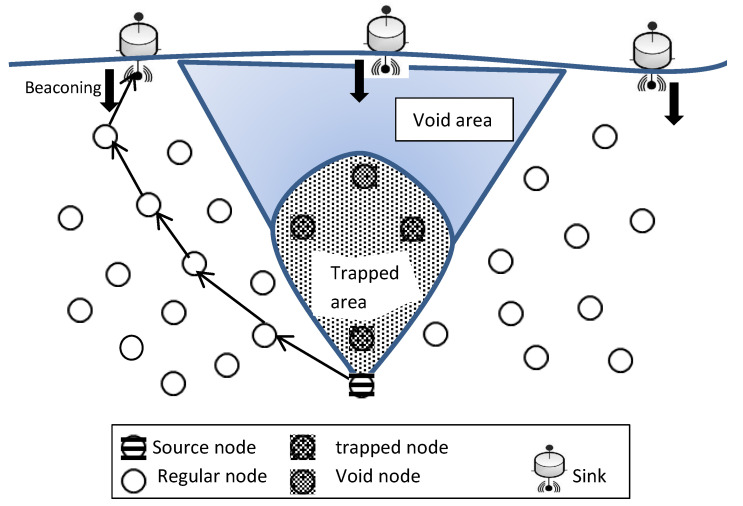
VAPR voids and trapped areas.

**Figure 7 sensors-22-09525-f007:**
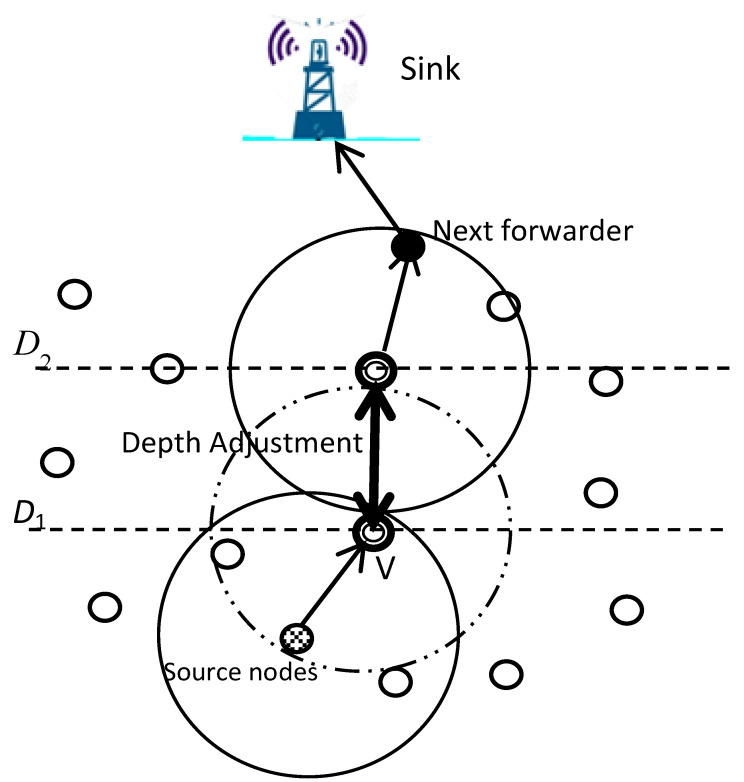
Depth Adjustment.

**Figure 8 sensors-22-09525-f008:**
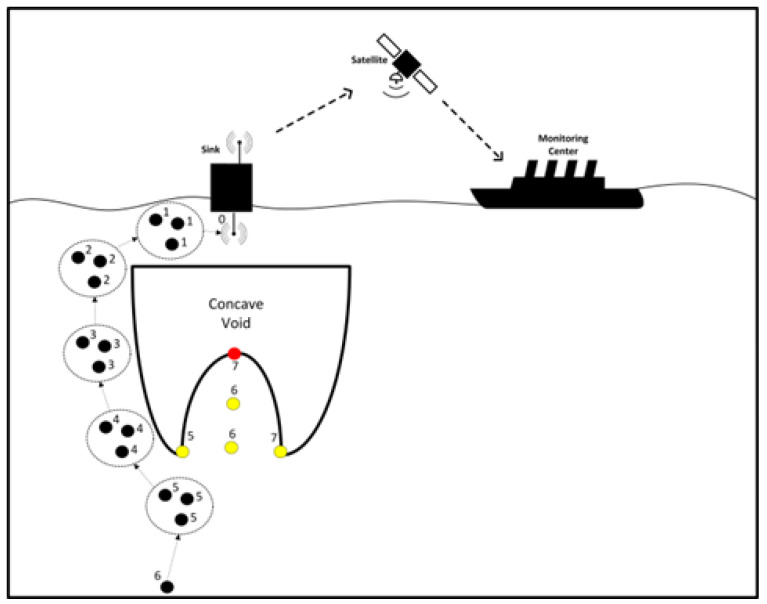
Void-handling technique [46].

**Figure 9 sensors-22-09525-f009:**
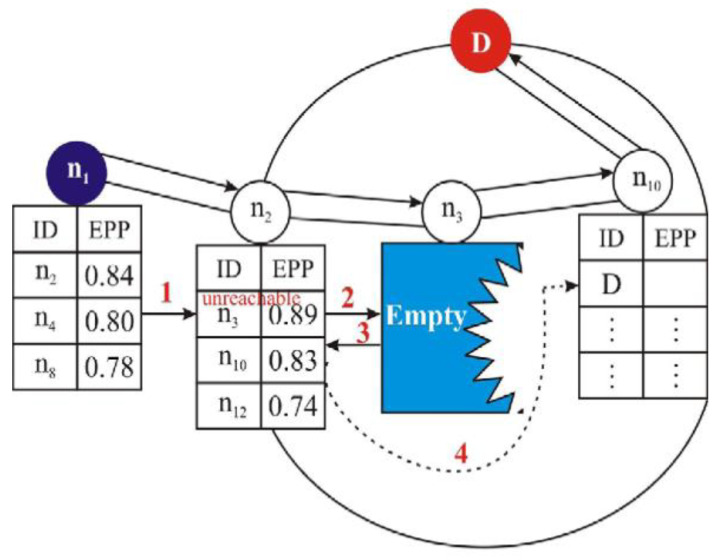
VHGOR recovery mode [48].

**Figure 10 sensors-22-09525-f010:**
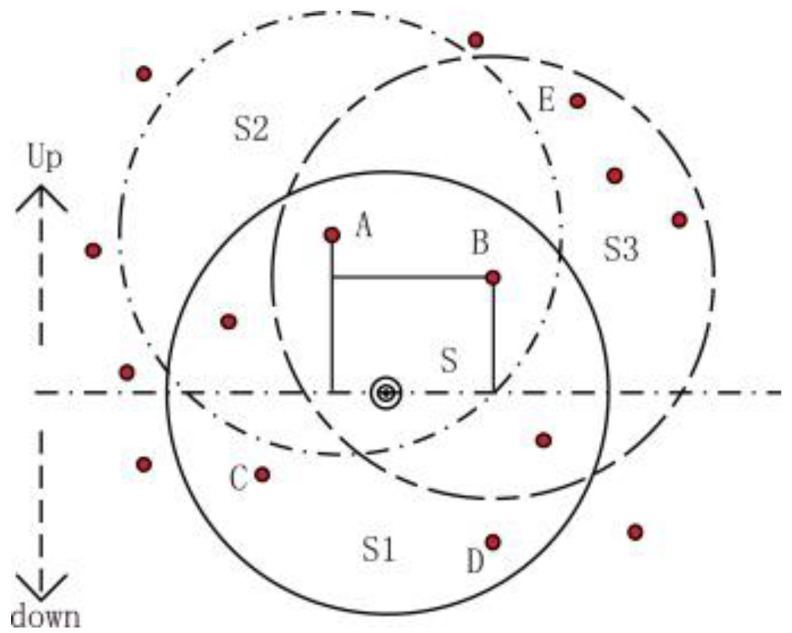
Void area problem [49].

**Figure 11 sensors-22-09525-f011:**
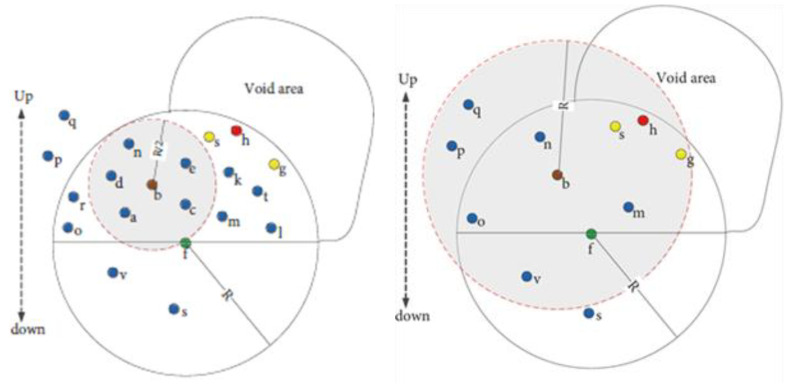
Resizing the forwarding area, sparse density on the right and dense density on the left [50].

**Figure 12 sensors-22-09525-f012:**
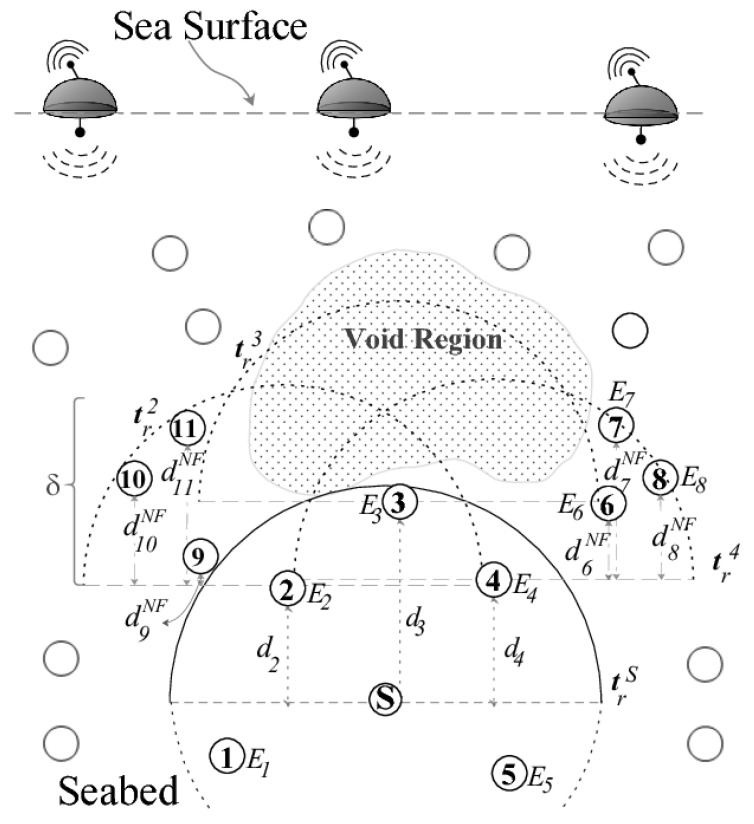
Holding time calculation parameters [52].

**Figure 13 sensors-22-09525-f013:**
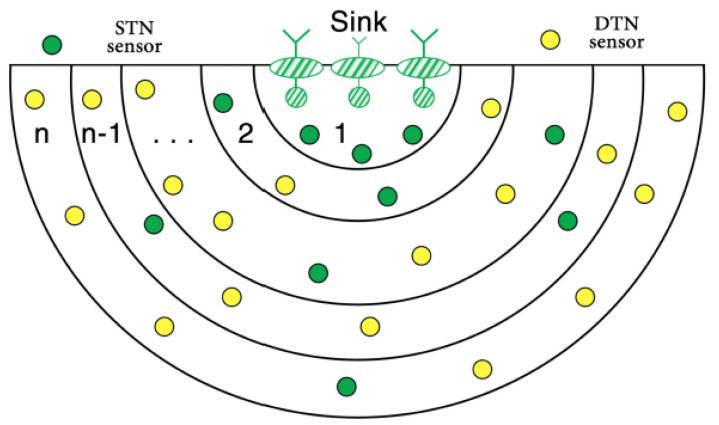
Layering structure in TORA at the node localization phase [53].

**Figure 14 sensors-22-09525-f014:**
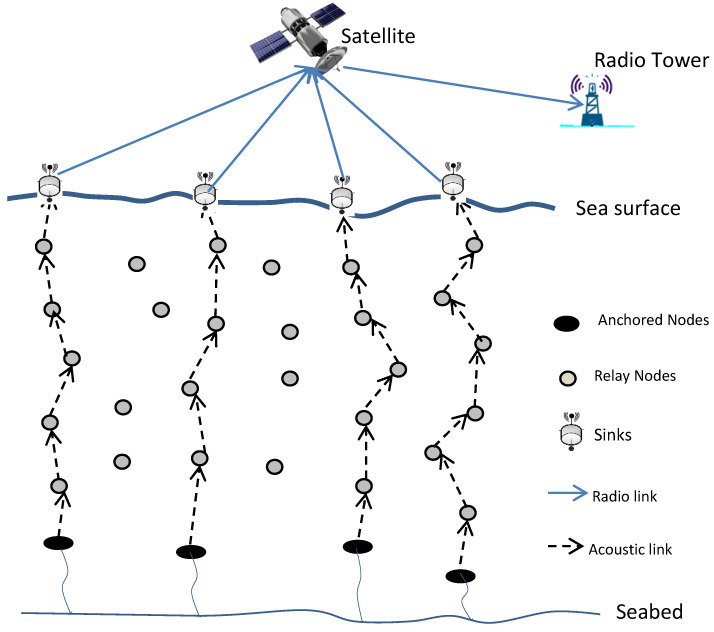
EBER^2^ Network Topology.

**Figure 15 sensors-22-09525-f015:**
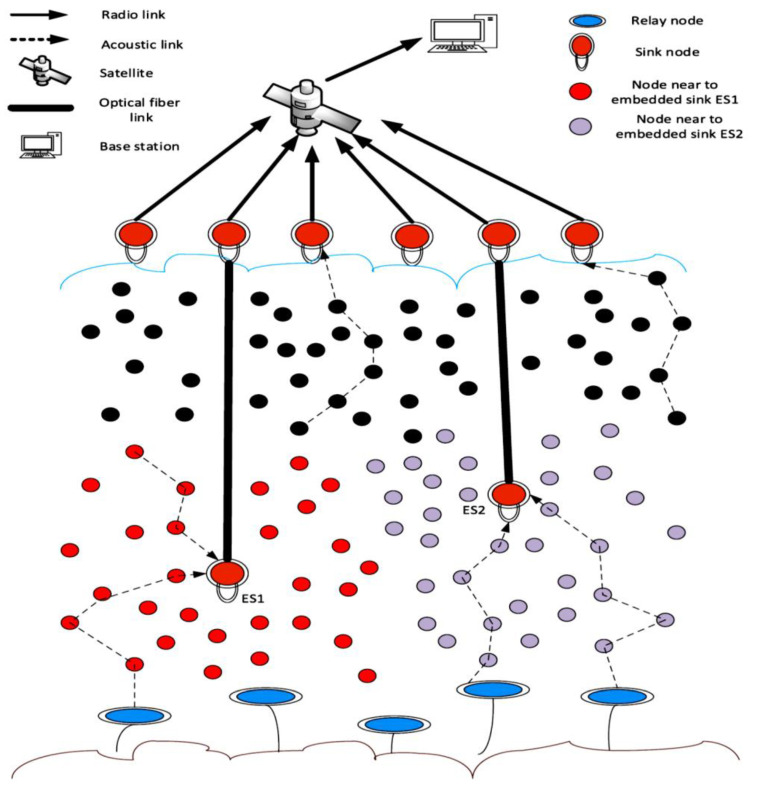
Network Topology with Two Embedded Sinks [54].

**Figure 16 sensors-22-09525-f016:**
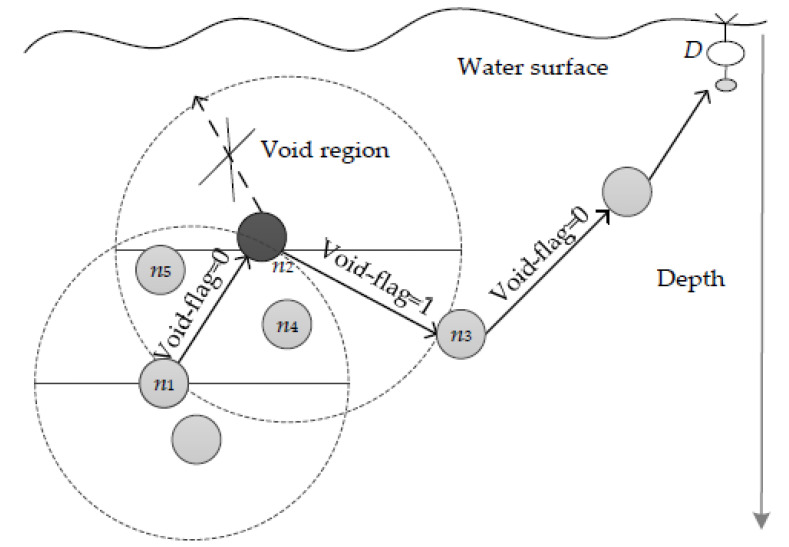
EDORQ void recovery mode [55].

**Figure 17 sensors-22-09525-f017:**
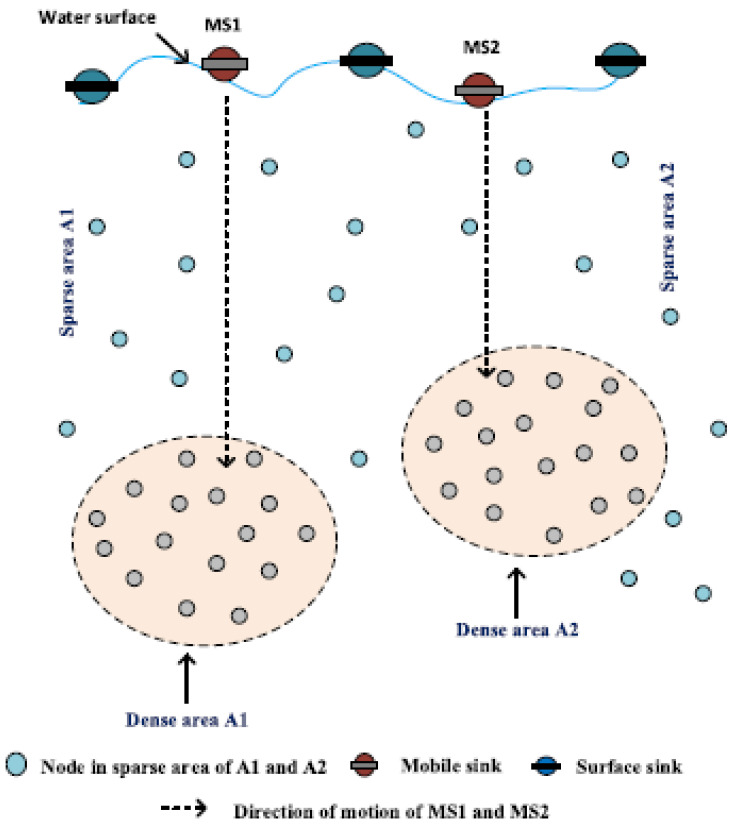
Sink mobility model [56].

**Figure 18 sensors-22-09525-f018:**
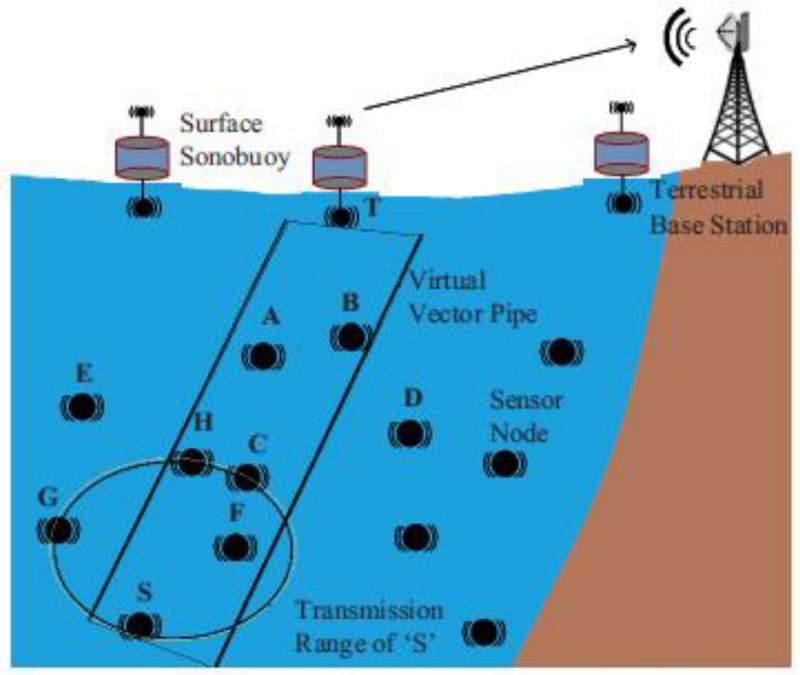
**The** best forwarder selection process in the SEEORVA protocol [58].

**Figure 19 sensors-22-09525-f019:**
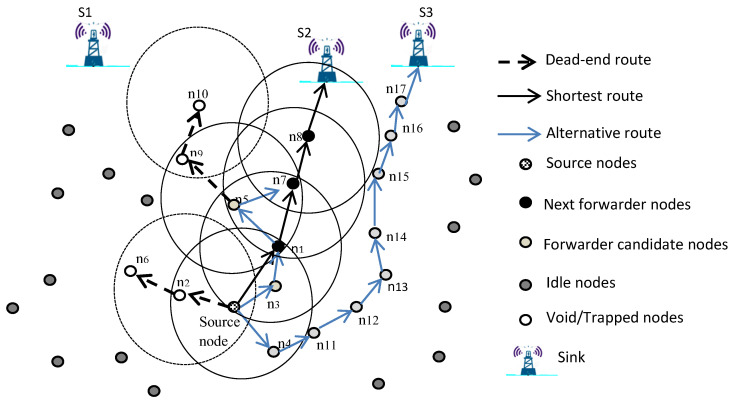
Underwater Network Architecture Model of the EEDOR-VA Protocol.

**Table 1 sensors-22-09525-t001:** Comparison between void avoiding OR protocols.

Protocol	Category	Forwarding Set Selection Category	Sink(s)	Requirements	Knowledge Required/Maintained	Advantage	Disadvantage
HydroCast [43]	Pressure-based routing	Sender-side	Multi-sink	Nodes with special H/W	2-hop connectivity and the pairwise distances for the neighboring nodes	Reduce end-to-end delay. High delivery ratio. Void handling technique by using recovery path.	High energy consumption due to repeating the process of finding a detour path. High overhead due to requiring 2-hop neighboring nodes information.
VAPR [44]	Geography-based routing	Sender-side	Multi-sink	SEA Swarm nodes	next-hop direction and hop distanceinformation at each node	Reduce end-to-end delay. Void handling technique by using directional opportunistic data forwarding algorithm. Use multi-sink reduces the sensor node’s battery drain and high traffic.	High energy consumption because utilizing enhanced beaconing and measuring the distance to the neighboring nodes and broadcasting of the measured information.
GEDAR [45]	Geography-based routing	Sender-side	Multi-sink	Nodes with special H/W	Position information of its own, neighbors and sink	Network topology control technique increases the connectivity of the network. Reduce the number of packet retransmissions. Void handling technique by utilizing a network topology control method.	High physical energy consumption due to node movement to adjust their depth. Ignore considering the sensor node energy level when selecting the forwarder node with high physical energy consumption may lead the protocol to be unable to select a forwarding node after a period of time due to exhausting their energy in physical movement.
IVAR [46]	Pressure-based routing	Receiver-side	Single-sink FIXED	Relay nodes and anchored nodes	Own depth, 1-hop neighbors and sink location	Eliminates all the routes leading to a void area and therefore no need for a switch to recovery mode	Redundant packet transmissions due to a hidden node problem. Redundant packet transmissions increases energy consumption.
OVAR [47]	Pressure-based routing	Sender-side	Single-sink FIXED	Relay nodes and anchored nodes	Own depth, 1-hop neighbors and sink location info.	No need for high overhead recovery mode for void handling since it ignores all the routes leading to a void area. Hidden node problem addressed by selecting the candidate nodes near each other. There is a trade-off between reliability and energy consumption.	Modifying the number of nodes of the forwarding set affected the reliability of the network.
VHGOR [48]	Geography- based routing	Sender-side	Single-sink	Geo. location is available	Own location/ neighboring table	Void node handled in two ways (i) convex void handling and (ii) concave void handling (or) recovery mode.	Consume restricted resources (memory through maintaining neighboring table, energy through nodes beacon).
WDFAD-DBR [49]	Pressure-based routing	Receiver-side	Multi-sinks	Anchored, relay and sink nodes.	Owen depth, 1-hop neighbor’s information and 2-hop neighbor’s depth.	Duplicated packets were handled by dividing the forwarding area and neighbor node prediction mechanism which help to reduce energy consumption. Sticking in void holes was reduced by using the depth of expected next hop.	Broadcast control packets and ACKs periodically consume the node’s battery and memory. Retransmission is required if the best forwarding node failed to transmit the packet. The flexibility of routing might be affected due to choosing a fixed primary forwarding area to form the forwarding set. The void area is not handled completely since the trapped nodes are not eliminated from the forwarding set.
EVA-DBR [50] andSORP [51]	Pressure-based routing	Sender-side	Multi-sinks	Anchored, relay and sink nodes.	Owen depth, 1-hop neighbor’s information and 2-hop neighbor’s depth.	By resizing the forwarding area the hidden problem is addressed in some cases. A trade-off between the energy consumption and latency based on the predefined maximum delay. Detect the void and trapped nodes before the data packet gets stuck in a void node.	Periodically broadcasting neighbor’s information consumes the node’s resources. Duplicated packets transmissions in spares network. Hidden problem may appear if the forwarding range chosen to be more than half of the transmission range.
EDOVE [52]	Pressure-based routing	Receiver-side	Multi-sinks	Anchored, relay and sink nodes.	Owen depth, 1-hop neighbor’s information and 2-hop neighbor’s depth.	Considering energy level as one of its parameters helps in reducing energy consumption and avoid energy holes.	Exchange the neighbor’s info, and maintains the neighbor’s table consumes the nodes resources. Duplicated packet transmissions increase the consumed energy. The void area is not handled completely since the protocol only addressed energy void holes.
TORA [53]	Geography- based routing	Receiver-based	Multi-sinks	Sink nodes and ordinary nodes.	Sinks and ordinary position information.	2-hop Ack is used to improve data delivery ratios and handle the void node issue. To reduce end-to-end delay and retransmission zero Ack is utilized.	High-energy consumption due to periodic hello and Ack messages. Extra cost due to equipping the ordinary nodes with a pressure sensor even though it is a geographic-based protocol.
EBER^2^ [54]	Pressure-based routing	Sender-side	Multi-sink	Anchored, relay and underwater sink nodes.	Two-hop Potential Forwarding nodes.	Residual energy of the nodes is used to reduce the duplicated packets and decreases the energy consumption. Transmission energy adaptability supports reducing the void holes. Embedded sinks used to increase the packet delivery ratio.	Suffers from large end-to-end delay as well as accumulative propagation distance. Communication between embedded sinks and on-surface sinks is costly. Duplicate packets to surface due to the node’s control power mechanism near the sinks
EDORQ [55]	Pressure-based routing	Receiver-side	Multi-sink	Relay nodes and anchored nodes	Own depth, 1-hop neighbors and sink location	Minimizes the total energy consumption and achieves a high packet delivery ratio. Handles the void area problem by switching to the void recovery mode. Suppresses the duplicate packet transmission through assigning a holding time to each candidate based on its Q-value	Recorded extra packet delay due to the holding time calculated based on Q-value. Switching to void recovery mode because void nodes decreases network performance by increasing packet delay and energy consumption. The protocol suffers from trapped nodes, which are not eliminated from the forwarding set. High computational cost.
RPSOR [56]	Pressure-based routing	Receiver-side	Multi-sink	Mobile sinks, relay nodes and anchored nodes	Owen depth, 1-hop neighbor’s information and 2-hop neighbor’s depth	Guarantee utilizing the shorts path. Prevents/reduces void hole formation in the network. Reduces duplicate packets	Maintenance tables exhaust node resources. Hello messages for exchanging global information increase the network overhead. Data transmission delay. Suffers from packet retransmission.
PCR [57]	Geography-based routing	Sender-side	Multi-sink	Nodes with power control mechanism.	Position information of its own, neighbors and sinks	Joint design of OR and power control to improve the link quality at each hop. Reduce the number of packet transmissions in the dense networks by reducing the transmission power level. Void handling technique by exploiting a power control mechanism.	Power control mechanise consumes more energy in forwarding set selection phase. Communication overhead due to broadcasting the beacon messages using different power levels.
SEEORVA [58]	Geography-based routing	Sender-side	Multi-sink	Underwater sensor nodes (source/ relay).	Nodes located in the virtual vector pipe between source node and sink	It uses energy threshold for energy conservation. Handle void areas by sending a data packet void_alert to the previous node. For security purposes, the protocol encrypts data packets before sending them through the network.	Large amounts of energy consumed to make nodes’ information global in the network In sparse networks, it is possible that the pipe does not have sufficient nodes to forward messages. The node density influences the pipe efficiency.
EEDOR-VA [59]	Pressure-based routing	Hybrid technique	Multi-sink	Nodes with special H/W (depth sensor).	Nodes’ hop count and depth.The previous HCREQ sender to unicast the HCREP to that node.	It proposed novel a technique to handle the void area problem. It identifies all trapped and void nodes. High PDR with energy efficiency maintaining. Minimizes redundant and retransmissions. Establishes a loop-free route between source and sink(s).	End-to-end delay due to route establishing. The protocol suffers from hidden terminal problem.

## Data Availability

Not applicable.

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
