# Peer review of "Void Avoiding Opportunistic Routing Protocols for Underwater Wireless Sensor Networks: A Survey"

_sensors, 2022, doi:10.3390/s22239525_

Round 1
Reviewer 1 Report
The author provided a comprehensive overview of the state-of-the-art of routing protocols for UWSNs that focuses on addressing the void area problem by utilizing OR. The paper has limited contributions and is somehow well-organized. I have some comment and suggestions as follow:
- The paper needs to be reorganized in terms of subsections and points.
- The authors need to add more references, especially recent references. The comparison studies are not considered state-of-the-art. Please add more recent works.
- Please add an abbreviation list for simplicity.
- How do Underwater Wireless Sensor Networks (UWSNs) benefit from the next-generation networks? Please refer to "6G Mobile Communication Technology: Requirements, Targets, Applications, Challenges, Advantages, and Opportunities" and "Survey on Multi-Path Routing Protocols of Underwater Wireless Sensor Networks: Advancement and Applications"
Reviewer 2 Report
This paper presented the aspects of routing protocols for UWSNs covering the main challenges facing researchers when designing routing protocols, the concept of void area problem in UWSNs and reasons behind this problem.
The structure is good. However, some works can be improved:
1. As a survey, the Introduction part is too simple. Please add some new works.
2. Figure 1 is so confused, please explain clearly.
3. why OR is most important in this paper.
4. many works are not news, authors may add some works
Round 2
Reviewer 1 Report
Thanks for addressing my comments. I do not have any further comments.
Reviewer 2 Report
This paper can be accepted